# Predictive Value of the Third Ventricle Width for Neurological Status in Multiple Sclerosis

**DOI:** 10.3390/jcm11102841

**Published:** 2022-05-18

**Authors:** Wojciech Guenter, Ewa Betscher, Robert Bonek

**Affiliations:** 1Department of Clinical Neuropsychology, Nicolaus Copernicus University, 87-100 Toruń, Poland, and Collegium Medicum, 85-094 Bydgoszcz, Poland; 2Division of Neurology and Clinical Neuroimmunology, Regional Specialized Hospital in Grudziądz, 86-300 Grudziądz, Poland; ewabetscher@gmail.com (E.B.); bonek.robert@gmail.com (R.B.)

**Keywords:** multiple sclerosis, cognitive impairment, EDSS, magnetic resonance imaging, the width of the third ventricle

## Abstract

The third ventricle width (3VW) is an easily calculated measure of brain atrophy. The aim of this study was to evaluate the relation of 3VW to cognitive impairment with adjustment for demographic and clinical confounders, including depression, anxiety, and fatigue, as well as to disability in patients with multiple sclerosis (MS). Symbol Digit Modalities Test, California Verbal Learning Test, Brief Visuospatial Memory Test-Revised, Expanded Disability Status Scale (EDSS), Hospital Anxiety and Depression Scale, and Modified Fatigue Impact Scale (MFIS) were analysed in 93 patients with MS. Neuropsychological performance was compared to that of 150 healthy controls. Axial images from 3D FLAIR were used to measure 3VW. In total, 25% of MS patients were impaired in at least two neuropsychological tests. Cognitive impairment and EDSS were associated with 3VW. Age and 3VW were the strongest predictors of cognitive impairment. The multiple regression model including age, 3VW, education, EDSS, and MFIS explained 63% of the variance of neuropsychological tests results, whereas 3VW, age and duration of the disease were significant predictors of EDSS. This study confirms the predictive value of 3VW for neurological status of patients with MS, especially for cognitive impairment after adjustment for demographic and clinical confounders.

## 1. Introduction

The pathophysiological basis of cognitive impairment and disability in multiple sclerosis (MS) is complex. Certainly, neurodegenerative changes accumulation is associated with progression of clinical symptoms. Many magnetic resonance imaging (MRI) outcomes were shown to be associated with the severity of disability and cognitive impairment, including lesion volume, cortical lesions, whole brain atrophy, cortical and deep grey matter atrophy, as well as damage to the normal-appearing brain tissue assessed with advanced MRI techniques [1,2,3,4,5,6,7,8].

Since implementation of automated tools developed for measuring brain volume loss in the clinical routine is difficult, the interest in simple methods of assessing brain atrophy increased. Two-dimensional measurements including the third ventricle width (3VW), the latral ventricle width, bicaudate ratio, and corpus callosum index correlate with whole brain volume and grey matter volume [9,10]. Furthermore, 3VW is an easily calculated measure performed manually once the MRI examination is finished. Measurement is not time consuming and could be implemented in the clinical practice. Moreover, 3VW was previously shown to be associated with cognitive deterioration and disability status of patients with MS [3,11,12,13]. Widening of the third ventricle is a marker of central atrophy corresponding with atrophy of thalamus and corpus callosum. Previous studies indicated that 3VW is a stronger predictor of cognitive disturbances than other measurements of atrophy in MS patients [3,14], and 3VW assessed with transcranial ultrasound (TCS) is a good predictor of cognitive decline in healthy persons as well [15]. Cognitive status might be affected by many demographic and clinical factors, including age, sex, level of education, disability, depression, and anxiety, as well as fatigue. Depression and anxiety affect up to 50% of patients with MS, whereas fatigue was noticed in 83% of individuals [16,17,18]; thus, it is important to include these in the analysis of associations between MRI measures and cognitive performance as a potential confounding factors.

The aim of the study was to assess whether 3VW is predictive for cognitive impairment and disability status in patients with MS. We especially aimed to evaluate the association between 3VW and cognitive performance with adjustments for demographic and clinical confounders, including depression, anxiety, and fatigue.

## 2. Methods

### 2.1. Participants and Procedures

Ninety three patients with MS, according to the revised 2010 McDonald’s criteria [19], and 150 healthy participants (HPs) were analyzed in the study. All of them participated in the validation study of the neuropsychological tools in Polish MS patients, which was conducted in the Department of Neurology and Clinical Neuroimmunology of Regional Specialist Hospital in Grudziądz, Poland, in 2017. Patients were recruited cross-sectionally to the validation study, with no selection for cognitive impairment and disability. Patients with other neurologic, psychiatric, or systemic diseases, drug or alcohol addictions, those who had an upper limb or visual impairment, or used a drug that would interfere with neuropsychological performance were excluded. All patients had been relapse-free and had not taken steroids for at least 1 month before assessment. A total of 39 patients (41.9%) were treated with DMT during the study, whereas 54 participants (58.1%) were not.

Physical and neurological examinations, including Expanded Disability Status Scale (EDSS), as well as an assessment of cognitive function, affective symptoms, and fatigue were performed on all participants. An MRI of the brain was performed within 5 days of the clinical evaluation in patients with MS. Seven MS patients who participated in the validation study were not included in the current analysis, as MRI was not conducted in these cases. The analyses presented in this paper are retrospective.

The experimental protocol of the validation study, which this paper is based on, was approved by the Bioethical Committee at the Regional Chamber of Physicians and Dentists in Bydgoszcz, Poland (No 39/2017, 19 September 2017). All methods were carried out in accordance with relevant guidelines and regulations. Informed consent was obtained from all patients who participated in the validation study of the neuropsychological tools in Polish MS patients, and who were then taken into the analysis in the current retrospective study.

### 2.2. Clinical and Neuropsychological Assessments

Neuropsychological examination included the following tests:Symbol Digit Modalities Test (SDMT) [20] is a simple substitution task evaluating attention and information processing speed. The written version of SDMT was administered. The test score was the number of correctly paired numbers with given geometric figures in 90 s.The Polish version of the California Verbal Learning Test (CVLT) [21] was used to assess verbal memory. The initial five learning trials were administered, and the test score was the total number of correct responses recorded across the five trials.Brief Visuospatial Memory Test-Revised (BVMT-R) [22] was used to evaluate visual memory. The initial three learning trials were administered, the test score was the sum of all three trials.

The Polish version of the Hospital Anxiety and Depression Scale (HADS) [23] was administered to assess depression and anxiety symptoms. The Polish version of Modified Fatigue Impact Scale (MFIS) [24] was used to evaluate fatigue.

### 2.3. MR Examination

Cranial MRI examinations were performed with a 1.5 T Philips Achieva scanner (Philips Medical Systems, Best, The Netherlands) in the Division of Radiology and Diagnostic Imaging at Regional Specialized Hospital in Grudziądz, Poland. The maximal slew rate was 80 mT/m/ms, with a maximum gradient strength of 33 mT/m. The brain MRI protocol included sagittal and axial T2-weighted turbo spin echo, axial T1-weighted spin echo, sagittal three-dimensional (3D) T1 fast field echo, sagittal 3D fluid-attenuated inversion recovery (3D FLAIR), and axial diffusion-weighted imaging. MR images were assessed by an experienced neuroradiologist and 3VW was measured as previously described [3]. Axial images from 3D FLAIR (TR, 4800 ms; TE, 315 ms; TF, 178; inversion time, 1660 ms; ST, 0.56 mm) were used for measurements. Slices were aligned to the lower borders of the corpus callosum. A line was drawn through the long axis of the third ventricle, parallel to the interhemispheric fissure in the section where the ventricle was most visible. The width was measured by drawing a second line perpendicular to the first at its midpoint and recording its lenght.

### 2.4. Statistical Analysis

Statistica 13.3 software (TIBCO Software Inc., Palo Alto, CA, USA) was used for statistical analyses. The normality of distribution of the variables was verified using the Shapiro–Wilk test. The homogeneity of variances in compared groups was evaluated with Levene’s test. The arithmetic means and standard deviations (SDs) are shown as measures of central tendency and dispersion for variables with normal distributions. Otherwise, the medians, 25–75th percentiles, and ranges are shown. Student’s *t*-tests were used to compare quantitative variables (with normal distributions and homogeneous variances). Mann–Whitney *U*-tests were used to compare two independent groups with quantitative variables that were not normally distributed. To compare more than two independent samples, the Kruskal–Wallis *H*-test was used. A chi-square test was used to compare nominal variables between two groups. Effect sizes were calculated with Cohen’s *d*. Correlations between two variables were assessed with the Spearman’s rank correlation coefficient (*R*), as variables were not normally distributed. Nonparametric partial Kendall regression was used to control for one confounding variable if variables were not normally distributed. A multiple regression model was used to calculate the effects of several predictors on a dependent variable. *p*-values of <0.05 were considered statistically significant.

## 3. Results

### 3.1. Clinical and Neuropsychological Outcomes

The main demographic characteristics of the groups and the clinical characteristics of patients with MS are detailed in Table 1. Three of the participants with MS did not perform HADS, and one single patient did not perform MFIS.

We compared the results of neuropsychological tests of MS patients to that of healthy subjects (Table 2). The multiple regressions were performed for each neuropsychological test with age, sex, and education as predictors of the test scores in the HPs group (Appendix A). Then, predicted scores were calculated for patients with MS. Predicted scores were subtracted from actual scores of patients for each neuropsychological test. If the difference was greater than standard deviation of the residuals of HPs, the score was classified as impaired. Specifically, impaired results of SDMT, CVLT, and BVMT-R were found in 33 (35%), 20 (22%), and 31 (33%) of patients with MS, respectively. Fifty-three subjects (57%) were impaired in at least one neuropsychological test, 23 patients (25%) in at least two tests, and 8 patients (9%) were impaired in all three cognitive tests. Patients who performed with impairment in at least two tests were classified as cognitively impaired (CI), whereas others were classified as cognitively preserved (CP).

Additionally, the standardised values of each cognitive test score were calculated. Then, a cognitive z-score was established for each MS patient, as a mean of the three standardised test scores obtained by the patient.

In the MS group the median value of the depression subscale of the HADS was 3.5 (25–75th percentiles, 1.0–7.0; range, 0–14.0). Seventeen participants (18.9%) had a score above 7 points. The median value of anxiety subscale of the HADS was 6 (25–75th percentiles, 4–8; range, 0–18). Thirty four patients (37.8%) had score above 7 points. The median value of MFIS was 34.5 (25–75th percentiles, 22.5–46.5; range, 1.0–71.0)

### 3.2. Relations of 3VW to Clinical and Neuropsychological Outcomes

The median value of 3VW was 5.5 mm (25–75th percentiles, 3.8–7.4; range, 1.1–13.5). When compared, 3VW measured in patients with MS in this study with age-dependent normal values obtained with similar methodology [25], taking the mean ± 1.5 SD as a normal range, 3VW was enlarged in 43 patients (46%).

Furthermore, 3VW was correlated with patients age (*R* = 0.35; *p* = 0.001) and sex, specifically males, who had significantly greater 3VW than females (median, 7.3; 25–75th percentiles, 5.5–8.2 vs. 4.6; 3.5–7.0; Mann–Whitney *U* = 497; *p* = 0.001). Duration of the disease was also associated with 3VW (*R* = 0.33; *p* = 0.001). The differences of 3VW among three clinical courses of MS were noticed (Kruskal Wallis *H* = 10.6, *p* = 0.005). The median value of 3VW was 4.3 (25–75th percentiles, 3.5–7.1) in RRMS, 6.5 (5.6–8.1) in SPMS, and 6.75 (3.40–9.35) in PPMS. The post-hoc analysis revealed the significant difference of 3VW between RRMS and SPMS patients (*p* = 0.004). After adjustment for age or disease duration the difference remained significant. No difference was observed between RRMS and PPMS, as well as between SPMS and PPMS. Disability assessed with EDSS was also related to 3VW (*R* = 0.36; *p* < 0.001).

In addition, 3VW was associated with the neuropsychological performance of patients with MS. We found strong correlations between 3VW and both cognitive z-score (*R* = −0.55; *p* < 0.000001) and SDMT (*R* = −0.51; *p* < 0.000001), as well as moderate correlations with CVLT (*R* = −0.45; *p* < 0.00001) and BVMT-R (*R* = −0.46; *p* < 0.00001) (Figure 1A–E). CI patients had greater 3VW compared to CP patients (median, 7.8; 25–75th percentiles, 5.7–8.9 vs. 4.8; 3.7–6.6; Mann-Whitney *U* = 399, *p* = 0.0003) (Figure 1F).

MFIS, HADS-D, and HADS-A scores were not associated with 3VW.

When adopting age-dependent normal values of 3VW [25], in the CI subgroup, 74% of patients had the third ventricle enlarged and 26% had 3VW within normal range, whereas within CP subjects, 37% had enlargement of the third ventricle and 63% had 3VW within normal range (*p* = 0.003) (Figure 2A). EDSS was also higher in subjects with enlargement of the third ventricle (*p* = 0.04) (Figure 2B).

### 3.3. Predictive Value of 3VW for Cognitive Impairment and Disability—Multiple Regression Models

To assess whether the correlation between 3VW and cognitive performance will change if the clinical and demographic factors are included in the analysis, a multiple regression model was performed, where 3VW, age, sex, education, duration of the disease, EDSS, HADS-A, HADS-D, and MFIS were preliminarly included as predictors of cognitive z-scores. Backward, as well as forward stepwise regressions indicated 3VW, age, education, and EDSS are significant predictors of cognitive performance. The result of the analysis was also confirmed with best subsets regression. Multiple regression model explaining 63% of the variance of cognitive z-scores is shown in Table 3. The strongest predictor of cognitive z-scores were age and 3VW.

A multiple regression model for EDSS was performed as well, where 3VW, age, sex and duration of the disease were preliminarly included in the analysis. Backward, as well as forward stepwise regressions indicated 3VW, age, and duration of the disease are significant predictors of EDSS. These predictors were confirmed with best subsets regression. Multiple regression model explaining 27% of the variance of EDSS is presented in Table 4.

## 4. Discussion

Neurodegenerative processes are a major cause of the long-term disability accumulation in MS [3,4,5]. The thalamus is among the first brain regions to become atrophic [26]. The thalamus has connections with widespread areas of neocortex and subcortical structures mediating many brain functions, thus damage to the thalamic nuclei might result in a wide range of neurologic symptoms, including cognitive imapirment. Moreover, extensive connections of the thalamus make this structure vulnarable to atrophy caused by the focal and diffused brain pathology, as a retrograde consequention of axonal damage in the white matter [27,28].

The third ventricle is a cavity located in the midline, separating the right and left thalamus. Widening of the third ventricle might result from atrophy of adjacent structures, especially thalami. Actually, it was shown that 3VW is associated with thalamic volume, and is enlarged in patients with MS [12,28]. There are several approaches to monitor brain atrophy in patients with MS. However, these techniques are still not implemented in clinical routines. As it is an easily calculated measure, 3VW could be widely used in the daily clinical setting as a marker of neurodegenerative processes reflected by atrophy of the diencephalon. However, the predictive value of this parameter for the neurological status of patients with MS must first be confirmed. Data on age-dependent normal values of 3VW are available [25,29]. According to these data, 46% of patients with MS in this study had an enlarged third ventricle.

Cognitive impairment occurs in 40–70% of patients with MS and the pathological basis of cognitive signs is complex [30]. Many MRI findings were shown to be associated with cognitive deterioration in patients with MS [1,2,3,4,5,6]. Cognitive disturbances, including information processing speed, as well as verbal and visual memory deficiencies, were associated with 3VW in this study. The strongest correlation was found for SDMT, which evaluates information processing speed and attention. Relations of 3VW to SDMT, CVLT, and BVMT-R were shown previously [3,31]. Moreover, 3VW was a better predictor of psychomotor speed and verbal memory disturbances than brain, neocortical, gray, and white matter volumes [31]. In addition, 3VW explained more variance in cognitive status than FLAIR, as well as T1 hypointense lesion volumes [3]. Thalamic atrophy, which is directly associated with 3VW, was also shown to be related to cognitive impairment [27,32].

After adjustment for demographic and clinical confounders, including depression, anxiety, and fatigue, we found that 3VW, after age, accounted for most of the variance in predicting cognitive dysfunction. Education, EDSS, and fatigue were also associated with cognitive status and such observations were made previously [33,34,35]. A multiple regression model including age, 3VW, education, EDSS, and fatigue explained 63% of the variance of neuropsychological performance in this study. Depressive and anxiety symptoms did not correlate with cognitive impairment in this study. Data regarding the association between depression or anxiety and cognitive disturbances is not uniform. However, the lack of such relations together with the presence of correlation between fatigue and cognitive functioning was noticed previously [36].

We found a weak association between 3VW and physical disability assessed with EDSS. A weak to moderate association between 3VW and EDSS was observed previously [12]. Physical disabilities of MS patients correlates with MRI metrics, as with T1 hypointense lesion volume and gray matter volume [5], for example. Particularly, thalamic atrophy and damage to the thalamus assessed with diffusion tensor imaging were found to be associated with EDSS [37,38]. However, correlation of thalamic volume to physical disability was weaker than to cognitive function [27], and this corresponds with the current results, as an association between 3VW and neuropsychological performance was stronger than between 3VW and EDSS. Age and duration of the disease were also associated with EDSS, which is in line with previous observations [5].

Although depressive symptoms in patients with MS are associated with atrophy of cortical and subcortical grey matter, including the left thalamus [39], 3VW was not associated with depression [40], and the same was observed in this study. We did not observe a correlation between anxiety and 3VW either. Fatigue can affect up to 80% of patients with MS and is considered to be related to thalamus pathology, and severity of fatigue in patients with MS is associated with thalamus atrophy [41]. However, this study did not reveal the association between 3VW and fatigue.

Significant thalamic atrophy was found in SPMS compared to PPMS and RRMS [26]. Moreover, 3VW was greater in SPMS than RRMS [31], which was confirmed in our study. This association was independent of age and duration of the disease. We noticed that males had greater 3VW than females. It is in accordance with the finding that atrophy, including thalamic atrophy, is more pronounced in MS males than females [42].

Transcranial sonography (TCS) is an alternative method to assess the cerebral ventricular system, and 3VW measured with TCS was also correlated with physical disability and cognitive performance in MS patients [43].

An important limitation of 3VW measurement is the low test–retest reliability [3,44]. Intra- and interrater reproducibility was not evaluated in this study. Image acquisition quality and slice positioning might affect reproducibility. Thus the importance of slice alignment to anatomical landmarks (lower borders of corpus callosum) is highlighted [12]. A limitation of this study is a lack of other MRI measurements, such as grey and white matter volumes, and advanced MRI techniques including diffusion tensor imaging, magnetization transfer imaging, or proton magnetic resonance spectroscopy evaluating diffused brain pathology to compare the predictive value of 3 VW with other MRI metrics. Due to retrospective character of the study, there is lack of 3VW measurement in healthy controls. Furthermore, longitudinal assessments of 3VW, cognitive and disability status changes over time are not available.

## 5. Conclusions

The predictive value of 3VW for the neurological status of patients with MS has been indicated in the previous research. The current study confirms these observations with adjustments for demographic and clinical confounders, including depression, anxiety, and fatigue. We found that 3VW is especially predictive for cognitive impairment, whereas association between 3VW and physical disability is weaker.

## Figures and Tables

**Figure 1 jcm-11-02841-f001:**
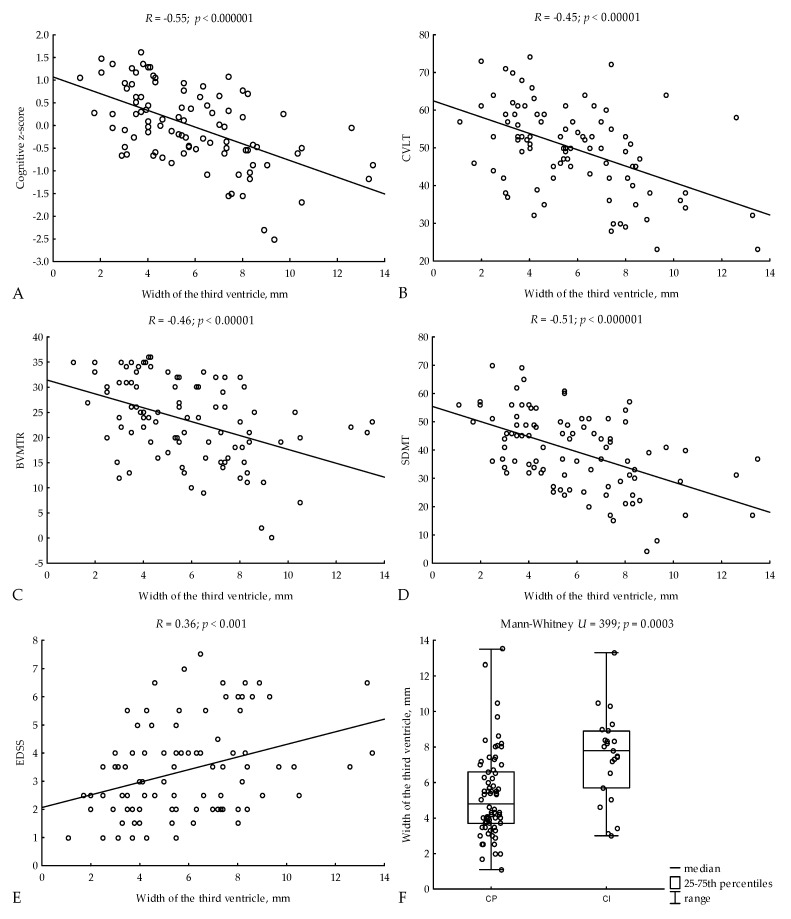
Relations of 3VW to cognitive z-scores (**A**), CVLT (**B**), BVMTR (**C**), SDMT (**D**), and EDSS (**E**). 3VW in CI and CP patients (**F**). R—Spearman’s rank correlation coefficient, CVLT—Californian Verbal Learning Test, BVMT-R—Brief Visuospatial Memory Test—Revised, SDMT—Symbol Digit Modalities Test, EDSS—Expanded Disability Status Scale, CP—cognitively impaired, CI—cognitively preserved.

**Figure 2 jcm-11-02841-f002:**
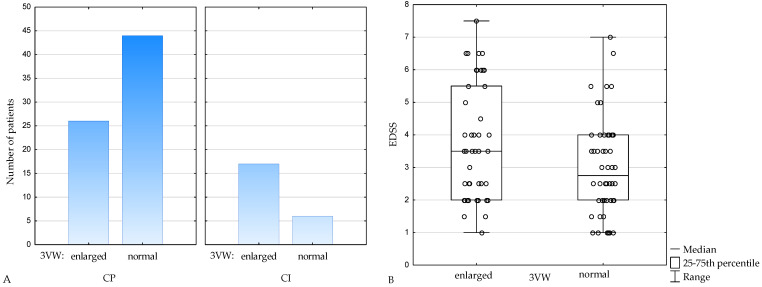
Number of patients with enlarged and normal 3VW in cognitively preserved (CP) and cognitively impaired (CI) subgroups (**A**), EDSS in patients with enlarged and normal 3VW (**B**). 3VW—the third ventricle width, CP—cognitively preserved, CI—cognitively impaired, EDSS— Expanded Disability Status Scale.

**Table 1 jcm-11-02841-t001:** Demographic and clinical characteristics of the study population.

Characteristic	MS Group *n* = 93	HPs Group *n* = 150	*p*
Age (years)			0.02
median (25–75th percentiles)	41 (33–50)	37 (29–48)	
range	20–67	18–64	
Female-to-male ratio	67:26	109:41	0.97
Education (years)			0.11
median (25–75th percentiles)	13 (12–17)	15 (12–17)	
range	8–18	8–21	
Disease duration (years)			
median (25–75th percentiles)	9 (4–16)		
range	0.1–44		
Disease course, *n* (%)			
RRMS	60 (64.5)		
SPMS	25 (26.9)		
PPMS	8 (8.60)		
EDSS			
median (25–75th percentiles)	3.0 (2.0–4.0)		
range	1.0–7.5		

MS—multiple sclerosis, HPs—healthy participants, RRMS—relapsing remitting multiple sclerosis, SPMS—secondary progressive multiple sclerosis, PPMS—primary progressive multiple sclerosis, EDSS—Expanded Disability Status Scale. Mann–Whitney *U*-tests were used to assess differences in age and education, whereas a chi-square test was used to assess female-to-male ratios.

**Table 2 jcm-11-02841-t002:** Neuropsychological tests results in MS and HPs groups.

	MS Group	HPs Group	Cohen’s *d*	*p*
SDMT, median	41	49	0.64	<0.0001
(25–75th percentiles)	(31–50)	(42–55)		
CVLT, mean ± SD	49.9 ± 11.5	54.1 ± 10.3	0.39	0.003
BVMT-R, median	24	29	0.48	0.003
(25–75th percentiles)	(18–30)	(24–32)		

MS—multiple sclerosis, HPs—healthy participants, SDMT—Symbol Digit Modalities Test, CVLT—Californian Verbal Learning Test, BVMT-R—Brief Visuospatial Memory Test—Revised, SD—standard deviation. Independent sample *t*-test was used for CVLT; Mann–Whitney *U*-tests were used for SDMT and BVMT-R.

**Table 3 jcm-11-02841-t003:** Multiple regression model for the cognitive z-scores.

F = 31.34; SE = 0.53; Corrected *R*^2^ = 0.63; *p* < 0.000001				
	β	SE_β	*t*	*p*
age	−0.32	0.08	−4.20	0.00006
3VW	−0.31	0.07	−4.24	0.00006
education	0.23	0.07	3.19	0.002
EDSS	−0.21	0.08	−2.70	0.008
MFIS	−0.16	0.07	−2.18	0.03

β—standardized beta coefficient, SE—standard error, *t*—*t* test statistic, 3VW—width of the third ventricle, EDSS—Expanded Disability Status Scale, MFIS—Modified Fatigue Impact Scale.

**Table 4 jcm-11-02841-t004:** Multiple regression model for EDSS.

F = 12.21; SE = 1.42; Corrected *R*^2^ = 0.27; *p* < 0.000001				
	β	SE_β	*t*	*p*
age	0.26	0.11	2.34	0.02
duration of the disease	0.24	0.11	2.25	0.03
3VW	0.22	0.09	2.29	0.02

β—standardized beta coefficient, SE—standard error, *t*—*t* test statistic, 3VW—width of the third ventricle.

## Data Availability

The data presented in this study are available on request from the corresponding author. The data are not publicly available due to privacy restrictions.

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
