# Peer review of "Predictive Value of the Third Ventricle Width for Neurological Status in Multiple Sclerosis"

_jcm, 2022, doi:10.3390/jcm11102841_

Round 1
Reviewer 1 Report
In their manuscript, “Predictive value of the third ventricle width for neurological status in multiple 2 sclerosis”, Guenter et al. investigated third ventricle width as potential predictive radiological biomarker for cognitive impairment in people with MS.
The Manuscript is of limited novelty (several studies have been performed on the theme) and presents serious methodological flaws.
The complete absence of analysis regarding other neuroradiological parameters (such as global and specific cortical atrophy, white matter lesion load, cortical lesions, presence of contrast medium capturing lesions), as well as the lack of an intra- and interrater reproducibility analysis for a measurement with a low test-retest reliability represent relevant weaknesses in a study that aims to identify a neuroradiological biomarker with a predictive role.
Author Response
Dear Reviewer,
Thank you for your comments and feedback regarding our research. There is lot of original papers and review articles concernig associations between neuroradiological parameters and disability / cognitive impairment, it is mentioned in the introduction section. The aim of this study was to assess if a simple two-dimensional MRI parameter with a set of clinical data can explain cognitive impairment and disability in patients with MS. Model including 3VW, age, education, EDSS and fatigue explained 63% of variance of neuropsychological performance, and this result is new. There was just one study assessing correlation between MRI markers including third ventricle width and cognitive impairment with adjustment for depression, anxiety and fatigue (Artemiadis et. al., 2018). Correlation od 3VW with EDSS was weak and this result is of limited novelty indeed.
Reviewer 2 Report
The manuscript entitled „Predictive value of the third ventricle width for neurological status in multiple sclerosis” is comparing the 3rd ventricule width (3VW) of 93 pwMS and 150 healthy controls and found a predictive value of the 3VW regarding cognitive impairment.
The manuscript is easy to follow and understand.
Minor points:
- The tables format could be improved. In table 3 and 4 the values of the 1st row are not explained eg. beta, SE_beta, t and P.
- Table 2 could be better visualized as boxplots.
- How were the data adjusted for age, as this parameter was significantly different?
- Were the MRI measurements standardized to the patients and healthy controls, eg. standardized timepoint of the MR imaging, hidration, menstrual cycle?
- Are there any other techniques in the literature regarding atrophy measurement eg. TCD?
Author Response
Dear Reviewer,
Thank you for your valuable comments and feedback regarding our research. We have made some changes to the manuscript, as well as recorded the answers to the comments below.
The tables format could be improved. In table 3 and 4 the values of the 1st row are not explained eg. beta, SE_beta, t and P.
Explanations of abbreviations were added.
Table 2 could be better visualized as boxplots.
It seems to us that Table 2 is clear, it does not contain a large amount of data. If it is necessary to change the table to boxplots it is possible to do.
How were the data adjusted for age, as this parameter was significantly different?
To compare neuropsychological tests results between patients with MS and healthy controls (Table 2) we used multiple regressions models including age. Age correlated with each cognitive test, however MS remained significant in these models (beta=0.25, p<0.0001 for SDMT, beta=0.2, p=0.0005 for BVMTR, beta=0.13, p=0.03 for CVLT).
As SDMT and BVMTR results in our groups were not characterised by normal distribution, we also checked age as a potential confounding variable with partial Kendall’s tau. In these calculations significant difference between MS and HC was proved as well (partial tau=0.23, p<0.0001 for SDMT, partial tau=0.16, p= 0.0002 for BVMTR).
In the multiple regressions used to classify participants as cognitively preserved or impaired, demographic factors were included (age, sex and education).
Were the MRI measurements standardized to the patients and healthy controls, eg. standardized timepoint of the MR imaging, hidration, menstrual cycle?
MRI was not performed in healthy controls. The MRI was performed within 5 days of the clinical and neuropsychological evaluation. Hidration and menstrual cycle was not assessed.
Are there any other techniques in the literature regarding atrophy measurement eg. TCD?
TCS is mentioned now.
Reviewer 3 Report
The authors present a retrospective analysis of 93 patients with multiple sclerosis (MS) compared to 150 healthy participants. In both groups, neuropsychological testings were performed and compared. Subjects were classified as cognitive impaired (CI) or cognitive preserved (CP), MS patients were clustered in 3 groups - RRMS – relapsing remitting multiple sclerosis, SPMS – secondary progressive multiple sclerosis, PPMS – primary progressive multiple sclerosis. In patients with MS an MRI of the brain was performed and width of third ventricle was measured. The width of third ventricle was compared to the results of the neuropsychological testing. Furthermore, a multiple regression analysis was performed, where age, third ventricle width, sex, education,duration of the disease, EDSS, HADS-A, HADS-D, and MFIS were preliminarly included as predictors of cognitive z-scores.
In general, the study is well structured with good statistical analysis. The cohort and the control group are large, the inclusion criteria are clear and the topic is of high relevance. But, and this is my main reservation on the study, there are even many publications in the current and previous literature addressing the identical issue. It has been published for many times, that in MS patients, an enlarged third ventricle is associated with cognitive impairment. So, my main question on this study is: what is really new and innovative compared to the literature? Or is the complete study a confirmation-study?
After I carefully reviewed this manuscript, I have a few reservations and recommendations for the authors:
- The title of the study focuses on the predictive value of the third ventricle width for neurological status in MS, so I think the authors should also focus more on the third ventricle in general. I miss information on normal, age-dependent values of the third ventricle. What is the diagnostic value of single third ventricle values in MS patients? Can measurement of the third ventricle be used as a screening tool for cognitive impairment? The differential diagnosis hydrocephalus should be kept in mind.
- The authors, moreover, write, that the third ventricle can easily be measured on MRI and this method can be implemented in the clinical practice. But the easiest way would be measurement of the third ventricle using trans-temporal ultrasound. Several studies have been published on this method especially for MS patients. I think this method should be mentioned as an important option.
- As I mentioned above, I think it is necessary to highlight what is new in this study compared to the current literature. This should be emphasized in the introduction as well as in the discussion part.
- As the authors wrote, a limitation of their study is, that MRI was only performed in MS patients. I also think it would be very interesting to compare the MRI results of MS patients to the healthy cohort. As the data are obviously not available, I would recommend to at least discuss the third ventricle values of MS patients with age-dependent normal values according to the literature. Serial investigations (MRI and cognitive testing) would be a very interesting and more innovative topic.
- The language/spelling of the manuscript needs some correction, as well. The font size differs within the script without any discernible reason (for example is the font size for the Statistical Analysis part larger), what makes it a bit messy. There are furthermore, some spelling mistakes - Line 131 “Than, predicted scores were calculated for patients with MS .”
- Caption of Table 3&4 Line 177&184 “SD – standard terror” , line 127 PPMS – primary progresive multiple sclerosis.
Author Response
Dear Reviewer,
Thank you for your valuable comments and feedback regarding our research. We have made some changes to the manuscript, as well as recorded the answers to the comments below.
The title of the study focuses on the predictive value of the third ventricle width for neurological status in MS, so I think the authors should also focus more on the third ventricle in general. I miss information on normal, age-dependent values of the third ventricle. What is the diagnostic value of single third ventricle values in MS patients? Can measurement of the third ventricle be used as a screening tool for cognitive impairment? The differential diagnosis hydrocephalus should be kept in mind.
Some information concerning third ventricle, especially references to normative data, were added.
The authors, moreover, write, that the third ventricle can easily be measured on MRI and this method can be implemented in the clinical practice. But the easiest way would be measurement of the third ventricle using trans-temporal ultrasound. Several studies have been published on this method especially for MS patients. I think this method should be mentioned as an important option.
TCS is mentioned now.
As I mentioned above, I think it is necessary to highlight what is new in this study compared to the current literature. This should be emphasized in the introduction as well as in the discussion part.
There was just one study assessing correlation between MRI markers including third ventricle width and cognitive impairment with adjustment for depression, anxiety and fatigue (Artemiadis et. al., 2018). Especially fatigue has not been included before, except for the study by Artemiadis . Moreover a lot of studies evaluating association between 3VW and cognitive function were performed on small groups (30-50 participants). We think that interesting and new result is regression model presented in the table 3 – model including a single two-dimensional parameter from MRI and clinical/demographic data explained 63% of variance of cognitive performance. These topics are highlighted in the discussion.
As the authors wrote, a limitation of their study is, that MRI was only performed in MS patients. I also think it would be very interesting to compare the MRI results of MS patients to the healthy cohort. As the data are obviously not available, I would recommend to at least discuss the third ventricle values of MS patients with age-dependent normal values according to the literature. Serial investigations (MRI and cognitive testing) would be a very interesting and more innovative topic.
Such analysis was added - 3VW of patients was compared with normal age-dependent values. We also analysed cognitive performance and EDSS in patients with normal and enlarged 3VW based on normal values (Figure 2).
The language/spelling of the manuscript needs some correction, as well. The font size differs within the script without any discernible reason (for example is the font size for the Statistical Analysis part larger), what makes it a bit messy. There are furthermore, some spelling mistakes - Line 131 “Than, predicted scores were calculated for patients with MS .”
- Caption of Table 3&4 Line 177&184 “SD – standard terror” , line 127 PPMS – primary progresive multiple sclerosis.
Corrected
Round 2
Reviewer 1 Report
I thank the authors and appreciate the effort made. However, I remain of the opinion that the study in the present form has methodological errors and severe limitations (absence of other MRI parameters, absence of intra- and inter-rater reproducibility evaluation, absence of standardized MRI protocols linked to low reliability of the measures, absence of controls) that do not allow to determine whether the third ventricle width acts as a dependent or independent factor (from other neuroradiological parameters) in relation to cognitive deficits. moreover, most studies are currently evaluating the role of the ultrasound study of the third ventricle width, so the novelty and potential impact of the study is very limited.
Reviewer 3 Report
The authors implemented nearly all of my recommendations- information on third ventricle normal values, diagnostic value of third ventricle width in MS and TCS of Third ventricle was added. A comparison of 3VW of MS patients vs. age-dependent normal values and an analysis of cognitive performance in patients with normal vs. enlarged 3WV was added as well. This information increased the quality of the article. The manuscript was also corrected regarding spelling and language, so I would recommend to publish this article.